# Introducing Adversarial Dropout in Generative Multi-Adversarial Networks

**Gonçalo Mordido, Haojin Yang & Christoph Meinel**
Hasso Plattner Institute
University of Potsdam
14482 Potsdam, Germany
{goncalo.mordido,haojin.yang,christoph.meinel}@hpi.de

## Abstract

We propose to extend the original generative adversarial networks (GANs) framework to multiple discriminators and omit, or dropout, the feedback of each discriminator with same probability at the end of each batch. Our approach forces the generator to not rely on a given discriminator to learn how to produce realistic looking samples, but, instead, on a dynamic ensemble of adversaries. This promotes variety of the generated samples, leading to a richer generator less prone to mode collapsing. We show preliminary results on MNIST and Fashion-MNIST that sustain our claims.

## 1 Introduction

Generative adversarial networks (Goodfellow et al. (2014)), or GANs, are a framework composed of two models - one generator ($G$) and one discriminator ($D$). $G$'s goal is to learn the true data distribution by trying to fool $D$ in classifying the produced samples as real samples. On the other hand, $D$'s job is to better distinguish real and fake samples. However, one of the main problems with GANs is mode collapse, where $G$ is able to fool $D$ by only producing data coming from the same data mode. This leads to similar looking generated samples, which is not the ideal situation since it means $G$ was only able to learn a small segment of the true data distribution. Hence, this is the main problem we plan to tackle with this work.

Our solution consists in applying adversarial dropout to multiple adversarial GANs. Dropout was introduced by Hinton et al. (2012) and has been a widely used method in neural networks to prevent overfitting ever since. It simply consists of omitting the output of some neurons with a probability $d$, or dropout rate. We reformulate this problem in our use case by applying dropout to the feedback of each $D$ at the end of each batch, forcing $G$ to rely on a dynamic ensemble of adversaries. This ultimately induces variety in $G$'s output, reducing mode collapse, since $G$ now has to satisfy the different possible subset of discriminators that may remain in the ensemble at every batch.

### 1.1 Related Work

Several works consider maintaining the same framework architecture and, instead, changing the single models' objective functions to promote variety (Arjovsky et al. (2017); Chau Lui et al. (2017); Li et al. (2017); Zhao et al. (2016); Thomas Unterthiner (2018); Metz et al. (2016)). However, these approaches can be seen as complementary work, since extending the described frameworks to multiple adversaries and applying adversarial dropout could be a further way of promoting variety in the generated samples.

Regarding using multiple discriminators to prevent mode collapse, Neyshabur et al. (2017) proposed to train a single generator against an array of discriminators that operate on a different low-dimensional projection of the data, whereas Durugkar et al. (2016) used a single $G$ that trains against several discriminators considering different levels of difficulty. However, both of these approaches constrain $D$'s architecture to promote variety. We argue this to be a limitation from an extendibility stand point, which does not exist in our approach.

## 2 GENERATIVE ADVERSARIAL NETWORKS

In their original setting (Goodfellow et al. (2014)), generative adversarial networks (GANs) consist of two models, a generator ($G$) and a discriminator ($D$) are trained together by playing a minimax game:

$$\min_G \max_D V(D, G) = \mathbb{E}_{x \sim p_r(x)}[\log D(x)] + \mathbb{E}_{z \sim p_z(z)}[\log(1 - D(G(z)))], \qquad (1)$$

where $p_z(z)$ is the noise distribution used to sample $G$'s input, whilst $G(z)$ represents its output, *i.e.,* a fake sample originated from mapping the input noise to the data space. Subsequently, $p_r(x)$ is the data distribution and $D(x)$ represents $D$'s output, *i.e.,* the probability $p$ of sample $x$ being from the training set.

Hence, whilst $D$'s tries to do a better job at distinguishing correctly between real and fake samples, $G$ tries to maximise the probability of its generated samples being considered as real by $D$. To do this, $D$ uses the ground truth to update its parameters $\theta_D$ with the gradient updates $\nabla_{\theta_D}$ that minimise its loss. On the other hand, to have a better chance in fooling $D$ in the future, $G$ tries to exploit $D$'s weaknesses by slightly changing its output using the gradient updates $\nabla_{\theta_{G_D}}$ that maximise $D$'s loss regarding the previously generated samples.

## 3 ADVERSARIAL DROPOUT

We propose to extend the original GANs framework to multiple discriminators and a single generator. Moreover, we try to eliminate the mode collapse problem by forcing $G$ to learn from and appease a dynamic ensemble of adversaries, encouraging $G$ to produce samples from a variety of modes. The dynamism of the ensemble is achieved by dropping out the feedback of each $D$ with a certain probability $d$ at the end of every batch. This means that $G$ will only consider the losses of the remaining discriminators when updating its parameters, by using the average of the updated weights according to each remaining discriminator's loss. The value function $V$ used for the minimax game is then modified to:

$$\min_G \max_{\{D_k\}} \sum_{i=k}^{K} V(D_k, G) = \sum_{i=k}^{K} \delta_k(\mathbb{E}_{x \sim p_r(x)}[\log D_k(x)] + \mathbb{E}_{z \sim p_z(z)}[\log(1 - D_k(G(z)))]), \quad (2)$$

where $\delta_k$ is a Bernoulli variable ($\delta_k \sim Bern(1 - d)$) and $\{D_k\}$ is the set of all discriminators. The gradients calculated from the loss of a given discriminator $D_k$ are only used for the calculation of $G$'s final gradient updates when $\delta_k = 1$, with $P(\delta_k = 1) = 1 - d$. It is important to note that when all discriminators in the ensemble are dropped out, we randomly pick one discriminator $D_j \in \{D_k\}$ and follow the original objective function presented in equation (1), using solely the gradient updates related to $D_j$ to update $G$. Thus, the final value function used is:

$$F(G, \{D_k\}) = \begin{cases} \min_G \max_{\{D_k\}} \sum_{i=k}^{K} V(D_k, G), & \text{if } \exists k : \delta_k = 1 \\ \min_G \max_{D_j} V(D_j, G), & \text{if } \forall k : \delta_k = 0, \\ & \text{for } j \in \{1, ..., k\} \end{cases} \qquad (3)$$

Since each discriminator trains independently, *i.e.,* is not aware of the existence of the others, no changes were made on their individual gradient updates. Moreover, each $D$ updates its parameters at the end of every batch, even if dropped out. Furthermore, to force each $D$ to specialise in a different part of the mode space, we split the batch amongst the different discriminators, meaning that each $D$ trains with different samples. This enables $G$ to receive a more complete feedback over the data space, easing it to learn the real data distribution.

## 4 EXPERIMENTS

We validated our approach using MNIST (LeCun & Cortes (2010)) and Fashion-MNIST (Xiao et al. (2017)), with examples of mode collapse on both of these datasets being shown in Figure 1. We show preliminary results demonstrating the effects of using different combinations of number of discriminators and dropout rates in Figure 2. The same model architectures were used both for both datasets and models. more specifically 4 fully-connected hidden layers, with LeakyReLu and Sigmoid activations used for all hidden units and output units, respectively.

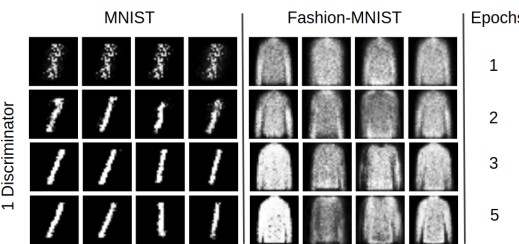

Figure 1: Mode collapse on MNIST and Fashion-MNIST using 1 discriminator. $G$ is only able to produce images of the number 1 (left) or jackets (right).

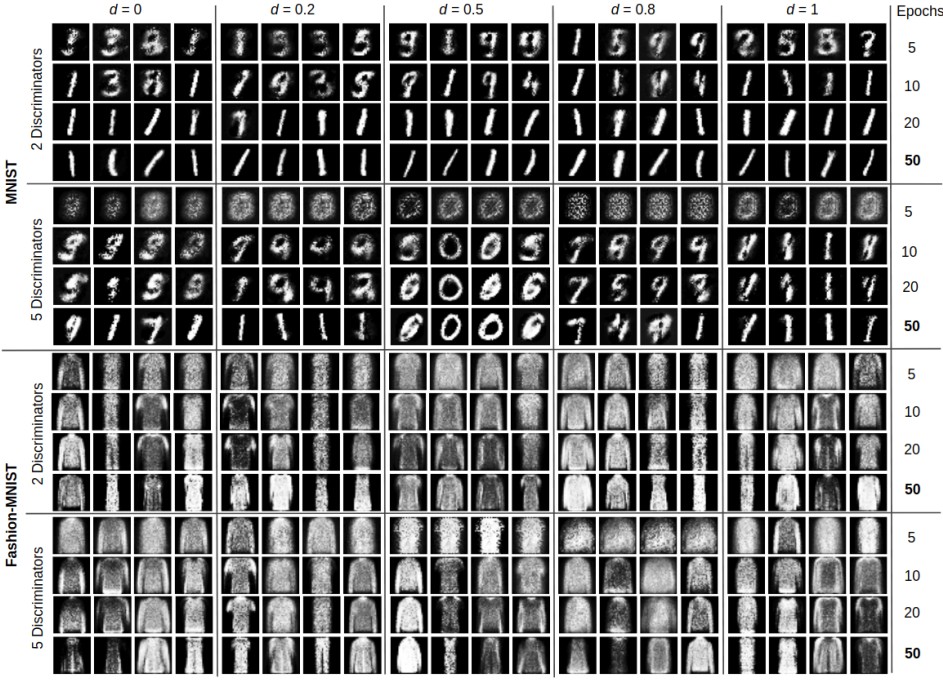

Figure 2: MNIST and Fashion-MNIST results. $G$ is now able to produce more diverse samples for a longer period of time. However, when using 5 discriminators, it is also visible that $G$ takes longer to learn, leading to a decrease in the quality of the generated samples in the earlier epochs. This is expected since $G$ has now access to more feedback and needs more time to handle it properly.

## 5 DISCUSSION AND FUTURE WORK

We show preliminary evidence that applying adversarial dropout promotes variety in the generated samples across time. In the future, we plan on testing the effects of using more discriminators and using more datasets. Moreover, we plan to use objective metrics which are suitable for detecting mode collapse, *e.g.*, FID (Heusel et al. (2017)), to further validate our solution.

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
