# OpenReview forum: "Introducing Adversarial Dropout in Generative Multi-Adversarial Networks"
_ICLR.cc/2018/Workshop — Reject_

### Official Review · AnonReviewer2 · 2018-02-22
**Highly incremental idea, unconvincingly explored, not validated quantitatively.**

**Rating:** 1
**Confidence:** 5

**Review:**

This abstract presents an incremental, trivial extension to GANs. The treatment of the problem is excessively informal, in particular the introduction. The name is also misleading, as one would expect something called "adversarial dropout" to drop components/terms/etc., well, adversarially. Instead, dropout is applied to terms of the loss function independently and at random (almost! see below).

Little attempt is made to formally introduce the problem being tackled. A three page abstract has limited space to convey results, and so one would expect it to be information dense. This one spends nearly half a page on background material that would be familiar to any mildly informed reader.

Equations 2 and 4 seem, under the most charitable assumption, to suggest iteratively *multiplying* parameters (elementwise, I suppose) by the gradient, with no mention of a step size/learning rate. I'm assuming this is a typo but it does not reflect well on the care taken with this submission.

Equation 3 introduces a set of Bernoulli gating variables on terms of a discriminator ensemble loss. We are told these are Bernoulli random variables, except that are not, because we are told that in the event that all of them are simultaneously zero, one of them is flipped to 1 uniformly at random. This presentation is confusing and seems quite arbitrary.

Equation 4 inexplicably adds a scaling term of 1/|{D_i'}| to the gradients (again, multiplying rather than subtracting, with no mention of a step size) that was not present in the loss function.

Zero details are given as to the architecture of the generator and discriminator in either the control or the condition. We are simply told that the baseline GAN exhibited mode collapse. Space constraints are a concern, but not overly: the presentation of the experiments done is a key part of the contribution and devoting more space to the details (or at least attaching an appendix).

The claimed improvements are based solely on qualitative evaluation against a very bad baseline (researchers successfully train GANs on MNIST quite frequently).

In summary: this work implements a trivial idea, lacks sufficient details to be reproduced, involves solely qualitative comparisons to a highly questionable baseline, contains multiple obvious errors in its equations.

---

> ### Public Comment · ~Goncalo_Mordido1 · 2018-03-21
> **Review response**
>
> Thank you for the valuable points brought up in your review! We updated the manuscript accordingly to meet some of your concerns.
>
> The issues noted in the equations with the gradient updates (previously Eq. 2 and 4) were indeed a typo and it was our lapse not noticing it a priori. In practice, we simply update the final weights using the average of the updated weights according to the loss of each discriminator in the set. Since the above equations do not really contribute to understanding our method, we removed them from the paper, and, instead, provided a clearer explanation on how the weights are really updated at the end of each batch. Moreover, we would like to highlight that the code implementation is correct and does not reflect the incorrectness showed in the aforementioned equations.
>
> Regarding the incompleteness of the presented value function (previously Eq. 3), we reckoned it would suffice to explain textually the special case of all discriminators being dropped out from the set. Nevertheless, we inserted a final value function that addresses this case in a more formal manner.
>
> Furthermore, the architectures used for both models and datasets can be found in the "Experiments" Section.
>
> Finally, and adding to the feedback from the other reviewers, we also agree that using objective metrics and more datasets is necessary to further validate the presented work as originally stated in the "Discussion and Future Work" Section of the paper.

---

### Official Review · AnonReviewer3 · 2018-03-09
**good idea, feels like it's been tried but cant place the reference**

**Rating:** 7
**Confidence:** 3

**Review:**

The paper proposes using dropout in the discriminator as a way to introduce discriminator ensembles. This is seen as a way to mitigate mode collapse. The idea is good, similarly motivated ideas have appeared in a few other papers, such as "mutual information regularizer" in "Improved GANs", and using multiple Generator / Discriminator pairs and cross-validating against the others: Jiwoong-Im et. al. (https://openreview.net/pdf?id=wVqzLo88YsG0qV7mtLq7).

The paper also formulates subtle details that one has to take care of wrt gradients of each batch when treating discriminator dropout as ensemble of discriminators.

Overall, the paper is well-written and well-motivated.

I feel like I've seen the "dropout in the discriminator" for GANs, but cant place a reference, so I'll ignore my feelings for review purposes.

---

### Official Review · AnonReviewer1 · 2018-03-10
**Discriminator Ensemble for GANs**

**Rating:** 5
**Confidence:** 4

**Review:**

The paper proposes to improve the robustness of GANs by training the generator against an ensemble of discriminators. While the approach has been previously explored in several papers (Durugkar et al. and Neyshabur et al. are cited) this paper explores a different approach to training the discriminator ensemble. Diversity is promoted between the discriminators via model level dropout and training on random sub-samples of the examples from the generator. This is a sensible line of work and the proposed approach is reasonable contribution to the space.

Currently the paper is lacking any quantitative metrics with the only results being visual comparisons of samples and identification of mode collapse. A concern is the baseline GAN model used in the paper is poor and mode collapses within a single epoch on MNIST and Fashion-MNIST. Studying and addressing mode collapse is an important problem in GAN research, but this paper only demonstrates that it addresses the problem in what is effectively a toy regime given the weak performance of the baseline. It's important to note that the original GAN paper shows a GAN trained on MNIST with much higher quality than the results using the techniques in this paper.

---

### Decision · Program_Chairs · 2018-03-20
**ICLR 2018 Workshop Acceptance Decision**

**Decision:**

Reject

**Comment:**

Based on the reviews, this paper has not been accepted for presentation at the ICLR workshop. However, the conversation and updates can continue to appear here on OpenReview.